# Endogenous Androgens Diminish Food Intake and Activation of Orexin A Neurons in Response to Reduced Glucose Availability in Male Rats

**DOI:** 10.3390/nu14061235

**Published:** 2022-03-15

**Authors:** Akira Takamata, Yuri Nishimura, Ayano Oka, Mayuna Nagata, Natsumi Kosugi, Sayaka Eguchi, Hiroko Negishi, Keiko Morimoto

**Affiliations:** 1Department of Environmental Health, Nara Women’s University, Kitauoya Nishimachi, Nara 630-8506, Japan; yn92@sussex.ac.uk (Y.N.); vaa_oka@cc.nara-wu.ac.jp (A.O.); vam_nagata@cc.nara-wu.ac.jp (M.N.); san_kosugi@cc.nara-wu.ac.jp (N.K.); sas_eguchi@cc.nara-wu.ac.jp (S.E.); ku178h.negishi@kyotokacho-u.ac.jp (H.N.); kei-morimoto@mail.koka.ac.jp (K.M.); 2Sussex Neuroscience, School of Life Sciences, University of Sussex, Brighton BN1 9QG, UK; 3Department of Food and Nutrition, Kyoto Kacho University, 3-456 Rinka-cho, Higashiyama-ku, Kyoto 605-0062, Japan; 4Department of Health and Nutrition, Faculty of Health Science, Kyoto Koka Women’s University, 38 Kadono-cho, Nishikyogoku, Ukyo-ku, Kyoto 615-0882, Japan

**Keywords:** orchiectomy, feeding behavior, testosterone, orexin, lateral hypothalamic area, perifornical area, arcuate nucleus, glucoprivation, glucose availability, 2-deoxy-d-glucose

## Abstract

Sex steroids modify feeding behavior and body weight regulation, and androgen reportedly augments food intake and body weight gain. To elucidate the role of endogenous androgens in the feeding regulation induced by reduced glucose availability, we examined the effect of gonadectomy (orchiectomy) on food intake and orexin A neuron’s activity in the lateral hypothalamic/perifornical area (LH/PFA) in response to reduced glucose availability (glucoprivation) induced by 2-deoxy-d-glucose (2DG) administration in male rats. Rats (7W) were bilaterally orchiectomized (ORX group) or sham operated (Sham group). Seventeen days after the surgery, food intake response to 2DG (400 mg/kg, i.v.) was measured for 4 h after the infusion. The same experiment was performed for the immunohistochemical examination of c-Fos-expressing orexin A neurons in the LH/PFA and c-Fos expression in the arcuate nucleus (Arc). Food intake induced by glucoprivation was greater in the ORX group than the Sham group, and the glucoprivation-induced food intake was inversely correlated with plasma testosterone concentration. Glucoprivation stimulated c-Fos expression of the orexin A neurons at the LH/PFA and c-Fos expression in the dorsomedial Arc. The number and percentage of c-Fos-expressing orexin A neurons in the LH/PFA and c-Fos expression in the dorsomedial Arc were significantly higher in the ORX group than the Sham group. This indicates that endogenous androgen, possibly testosterone, diminishes the food intake induced by reduced glucose availability, possibly via the attenuated activity of orexin A neuron in the LH/PFA and neurons in the dorsomedial Arc.

## 1. Introduction

Sex differences in feeding behavior and body weight regulation are well established in many animal species, and gonadal steroids are known to play roles in the regulation of body weight and feeding behavior [1,2,3]. In women, the prevalence of obesity increases after menopause, suggesting that ovarian hormones are possibly involved in the regulation of body weight [4,5,6]. A number of studies in animals have confirmed that estrogen, but not progesterone, exerts anti-obesity and anorexigenic actions [1,2]. On the other hand, the role of androgen in the regulation of feeding behavior and energy balance does not seem simple. It has been reported that andropause, or late-onset hypogonadism, may induce obesity and metabolic syndrome, and testosterone replacement may potentially be an effective treatment for weight management in obese men with hypogonadism possibly mediated via the androgen receptor (AR) [7,8,9]. It has also been reported in humans that, at physiologic levels, an association exists between higher levels of testosterone and favorable lean and fat measures [10], and testosterone dose-dependently reduced fat mass and increased lean body mass and body weight [11]. Thus, testosterone plays a role in the regulations of body composition and body weight in humans. However, few studies have been done on the effect of androgens on the regulation of eating behavior and its underlying mechanism due to methodological difficulties.

Animal studies have demonstrated that gonadectomy in male animals (orchiectomy) reduces and testosterone augments body weight gain, and androgen increases lean body mass and decreases fat mass [12]. On the other hand, the role of androgens in the regulation of food intake is controversial; some studies reported that androgen increased body weight gain with increased food intake, while other studies reported androgen increased body weight gain without altering daily food intake [1,13]. Thus, orchiectomy and androgen treatment seem to more reliably affect body weight and body composition than feeding behavior [1,13]. Additionally, a relatively high dose of exogenous testosterone attenuates food intake, which may be caused by the action of estradiol aromatized from testosterone [14].

The regulation of feeding behavior involves many regulatory mechanisms [15], and estrogens have been reported to act on various central feeding-related sites exerting anorexigenic responses [1,2,16]. It is well established that estrogens enhance satiety [17,18] or reduce meal size by enhancing the response to cholecystokinin, and estrogens enhance the neuronal activity of proopiomelanocortin (POMC) neurons, anorexigenic neurons, in the arcuate nucleus (Arc) [19]. In contrast, the role of testosterone in feeding behavior and its central mechanisms have been less studied.

In several studies, orchiectomy reduced meal number, increased meal size, and reduced daily food intake and body weight, suggesting that androgen may modify feeding behavior [20,21]. Orchiectomy also reportedly increased food intake and enhanced the inactivation of melanin-concentrating hormone (MCH) neuron response to glucose in fasting male rats, and testosterone replacement restored the enhanced inactivating response of MCH neurons to the plasma glucose level, suggesting that endogenous testosterone reduces satiety, which is induced by elevated postprandial plasma glucose concentration after fasting via the attenuation of MCH neurons’ response to glucose and increases food intake [21,22].

In our previous study, we demonstrated that intravenous 2-deoxy-d-glucose (2DG) administration-induced reduction in glucose availability (glucoprivation) and subcutaneous insulin-induced hypoglycemia stimulated feeding behavior and activated orexin A neurons and neurons in the lateral hypothalamic/perifornical area (LH/PAF) and neurons in the dorsomedial part of the Arc, probably neuropeptide Y (NPY) neurons, but did not activate MCH neurons [23]. This suggests that orexin A neurons in the LH/PAF and NPY neurons in the Arc are activated in response to reduced glucose availability; in other words, these neurons are involved in short-term feeding regulation [23,24]. Androgen receptors are relatively densely distributed in the lateral hypothalamic area [25], and the response of orexin A neurons to fasting was reportedly attenuated in male rats compared with the response in female rats, suggesting that male sex steroid possibly attenuated the activation of orexin A neurons in response to reduced glucose availability [26]. Feeding behavior is regulated by factors that stimulate and terminate feeding, and the integration of these regulatory functions determines food intake. As mentioned above, testosterone reportedly reduced satiety and increased food intake in fasting male rats. However, the role of endogenous androgens in the short-term regulation of feeding behavior and orexin A neuron activity remains unknown.

In the present study, to elucidate the role of endogenous androgens in the short-term regulation of feeding behavior, which is induced by reduced glucose availability, we examined the effect of orchiectomy on the feeding behavior and c-Fos expression, as a marker of neuronal activation [27,28,29], in the orexin A neurons in the LH/PFA in response to 2DG-induced glucoprivation, which specifically stimulates glucose-sensitive neurons.

## 2. Materials and Methods

All experiments were conducted in accordance with the guidelines for animal care and use of Nara Women’s University and with the Standards relating to the Care and Keeping and Reducing Pain of Laboratory Animals, the Ministry of the Environment, Japan. All of the experimental procedures employed in the present study were approved by the ethical committee for animal care and use of Nara Women’s University.

### 2.1. Animals and Surgery

Seven-week-old male Wistar rats (Jcr:Wistar; CLEA Japan, Japan) were assigned to two groups: rats were orchiectomized in one group (ORX group) and were sham operated in the other group (Sham group). The animals were housed in a chamber with an ambient temperature of 23 °C and a relative humidity of 40% under a 12-h/12-h light/dark cycle environment with lights on at 0700 and were given ad libitum access to standard rodent chow (CE-2, CLEA, Tokyo, Japan) and tap water before the experiments.

Rats were bilaterally orchiectomized under general anesthesia with pentobarbital sodium (50 mg/kg body weight; i.p.) and isoflurane (2%). Fourteen days after the orchiectomy, A polyvinyl chloride catheter (3 Fr polyvinyl chloride, ATOM, Saitama, Japan) was inserted into the right jugular vein, and the tip was placed at the superior vena cava/right atrium. The free end of the catheter was passed subcutaneously and exteriorized dorsally behind the neck through the midscapular incision. A miniature datalogger (Thermochron SL, KN Laboratory, Osaka, Japan) was implanted in the abdominal cavity to measure the intraabdominal temperature (T_abdo_) as an index of body core temperature. The patency of the venous catheter was maintained by flushing with heparinized isotonic saline (100 U/mL) every day.

Body weight was measured every day, and daily food intake and body weight were measured at 10:00 (Zeitgeber time; ZT3) for one week after the 7-day recovery from the orchiectomy/sham operation.

### 2.2. Response of Feeding Behavior to 2DG Administration

Four days after the catheterization surgery, we conducted the experiment to examine the food intake response induced by glucoprivation. The experiments were started at 10:00 (ZT3). Access to food was removed one hour before the experiment. A 0.2 mL blood sample was drawn through the venous catheter, and then 2DG (Sigma–Aldrich; 400 mg/kg body weight in 2 mL/kg body weight sterile distilled water; *n* = 7 in the ORX_2DG group and *n* = 6 in the Sham_2DG group) was infused through the venous catheter. To confirm that saline infusion did not activate feeding, the same volume of saline was infused in the control group (*n* = 4 in the ORX_saline group and *n* = 4 in the Sham_saline group). Access to food was provided immediately after 2DG administration, and blood sampling (0.2 mL) through the venous catheter and measurement of food intake were conducted every 60 min. T_abdo_ was also measured every 5 min throughout the experiments. The collected blood was immediately centrifuged, and plasma aliquots were stored at −20 °C until the plasma glucose concentration was measured.

The plasma glucose concentration, including exogenous 2DG, was measured with the mutarotase–glucose oxidase method (Glucose CII-test Wako; Wako Pure Chemicals, Osaka, Japan). Food intake was measured by manually weighing chow hoppers on a digital scale.

### 2.3. Response of Orexin Neurons in the LH/PFA and Neurons in the Arc to 2DG Administration

Three days after the experiment examining feeding behavior, we conducted experiments to immunohistochemically examine the c-Fos expression in the ORX-A neurons in the LH/PFA and also the c-Fos expression in the dorsomedial Arc. One hour after the removal of access to food, rats were intravenously injected with either 2DG or isotonic saline. In this experiment, access to water, but not food, was provided after the injection. Three hours after the injection, the rats were deeply anesthetized with pentobarbital. In the previous study and present study, the onset of feeding induced by 2DG injection occurred 1 h after the 2DG injection [23], and we set the timing for the examination of the expression of c-Fos 3 h after the 2DG/saline administration because the maximal c-Fos expression reportedly occurs approximately 1.5–2 h after stimulation [29]. Then, rats were transcardially perfused with ice-cold phosphate-buffered saline (PBS), followed by 4% paraformaldehyde in a 0.1 M phosphate buffer (pH 7.4) for fixation. The brain was removed and immersed in the fixative for more than two days at 4 °C. The fixed brain was immersed in 15% sucrose in PBS for one day and 25% sucrose in PBS for two days at 4 °C for cryoprotection. Frozen sections were cut coronally at a thickness of 30 µm with a cryostat microtome (LeicaCM3050S, Wetzlar, Germany).

### 2.4. Immunohistochemistry

To examine the neuronal activation of the orexin neurons induced by glucoprivation, we performed c-Fos and orexin A double staining. More details of procedures were reported previously [23]. Briefly, free-floating sections were preincubated with 5% normal goat serum and then incubated overnight at 4 °C with anti-c-Fos antibody (1:4000 dilution; Sc-52, Santa Cruz Bio-technology, Dallas, TX, USA) and with biotinylated secondary antibody (1:400 dilution; BA-1000, Vector, Burlingame, CA, USA) for 2 h, followed by the ABC Elite kit solution (1:400 dilution; Vector, CA, USA) for 2 h. Visualization was performed with 0.02% 3,3-diaminobenzidine (DAB) and 0.01 % H_2_O_2_ in a 50 mM Tris HCl buffer (pH 7.4). After the DAB reaction, the sections were incubated overnight at 4 °C with anti-orexin A antibody (1:2000 dilution; AB-3704; Chemicon, Darmstadt, Germany) and with secondary antibody conjugated with FITC (1:400 dilution; FI-1000, Vector, Burlingame, CA, USA) for 2 h. The sections were mounted on gelatin-coated glass slides and cover slipped with the mounting media for fluorescence (VECTASHIELD Hard Set; Vector, Burlingame, CA, USA). We also examined the c-Fos expression in the MCH neurons in the LH/PFA. In this experiment, we applied the same protocol as that for c-Fos and orexin A double staining, but an anti-MCH antibody (1:2000; H-070-47, Phoenix Pharmaceuticals, Inc., Burlingame, CA, USA) was used instead of the anti-orexin A antibody.

### 2.5. Data Analysis and Hormone Assay

Sections including the LH/PFA and Arc were identified using the rat brain stereotaxic atlas [30]. Three sections of every fifth section containing the LH/PFA and Arc (in between −2.56 and −3.60 mm from the Bregma) [30] were observed using a fluorescent microscope (Olympus BX-51, Tokyo, Japan), and bright-field images for c-Fos and fluorescence images for ORX-A neurons were obtained using a cooled-CCD camera (MicroPublishrer 5.0; QImaging, Surrey, BC, Canada), and these images were merged after the bright-field c-Fos images were converted into 8-bit images and inverted (ImageJ, NIH, Bethesda, MD, USA).

The number of neurons that expressed ORX-A and the number of ORX-A neurons that coexpressed c-Fos were counted unilaterally in the images (917 × 662 µm). The number of c-Fos immunoreactive nuclei in the Arc was also counted unilaterally. The percentage of the ORX-A neurons that coexpressed c-Fos was calculated.

The plasma testosterone concentration was measured with a commercially available enzyme immunoassay (EIA) kit (Cayman, Ann Arbor, MI, USA). The determination of testosterone in all samples was performed in a single assay.

### 2.6. Statistics

Data are shown as means with their standard errors (SE). Two-way analysis of variance (ANOVA) with repeated measures (one within and one between factors) followed by the Holm post hoc test was performed to determine the effects of orchiectomy (between factors) and time (within factors) on body weight change in all animals (*n* = 11 in each group) and 2DG-induced food intake and plasma glucose concentration in 2DG-infused rats (*n* = 7 in the ORX group; *n* = 6 in the Sham group). An unpaired Student’s t-test was performed to determine the effects of the orchiectomy on the body weight change, mean daily food intake, plasma testosterone concentration in all rats (*n* =11 in each group), and the percentage of c-Fos-ir nuclei in the ORX-A neurons at the LH/PFA, as well as on the number of c-Fos-ir nuclei in the Arc in 2DG-infused rats (*n* = 7 in the ORX group; *n* = 6 in the Sham group). Values of *p* < 0.05 were considered statistically significant.

## 3. Results

### 3.1. Plasma Testosterone Concentration, Daily Food Intake, and Body Weight Change

The plasma testosterone concentration was significantly lower in the ORX group than in the Sham group (Figure 1A).

The body weight before the orchiectomy/sham operation was not different between the groups, and the body weight two weeks after the orchiectomy/sham operation was lower in the ORX than in the Sham group (Figure 1B). Body weight gain at two weeks after the orchiectomy/sham operation was lower in the ORX group than the Sham group (Figure 1C). On the other hand, the mean daily food intake over one week was not different between the two groups (Figure 1D).

### 3.2. Response of Feeding Behavior to 2DG Administration

The administration of 2DG increased the plasma glucose concentration, including exogenous 2DG (Figure 2A), and also decreased T_abdo_ (Figure 2B), and these changes were not different between the ORX and Sham groups (Figure 2A,B). Glucoprivation induced by i.v. 2DG stimulated food intake in both the ORX and Sham groups, and food intake induced by glucoprivation was significantly greater in the ORX group than in the Sham group (Figure 2C).

Food intake stimulated by glucoprivation showed a significant inverse correlation with the plasma testosterone concentration (Figure 2D).

### 3.3. Hypothalamic Neuronal Response to 2DG Administration

Glucoprivation increased c-Fos expression in the orexin A neurons in the LH/PFA (Figure 3A,B). The percentage of c-Fos expression in the orexin A neurons induced by glucoprivation was greater in the ORX group than in the Sham group, while the number of orexin A neurons was not different among the groups (Figure 3C,D).

Glucoprivation induced by 2DG administration also increased c-Fos expression in the dorsomedial Arc (Figure 3E,F), and the number of c-Fos-expressing nuclei in the dorsomedial Arc after 2DG administration was greater in the ORX group than in the Sham group (Figure 3G). As in our previous study [23], 2DG-induced glucoprivation did not stimulate MCH neurons, and the percentage of c-Fos-expressing MCH neurons was not different among the groups (0.8 ± 0.2% in the ORX_2DG group; 0.1 ± 0.1% in the ORX_saline group; 0.7 ± 0.1% in the Sham_2DG group; and 0.1 ± 0.1% in the Sham_saline group).

## 4. Discussion

In the present study, we found that orchiectomy enhanced the food intake response to reduced glucose availability induced by intravenous 2DG administration, and glucoprivation-induced food intake was inversely correlated with the plasma testosterone concentration. We also found that the percentage of orexin A neurons that expressed c-Fos in the LH/PFA and the number of c-Fos-ir nuclei in the dorsomedial Arc were greater in the ORX group than in the Sham group. These findings indicate that endogenous androgens diminish the short-term feeding behavior induced by reduced glucose availability, or hypoglycemia, by attenuating the response of orexin A neurons in the LH/PFA and neurons in the Arc. Endogenous testosterone possibly plays a role in the attenuated short-term feeding behavior because food intake stimulated by glucoprivation was significantly inversely correlated with plasma testosterone level.

The response of orexin neurons to fasting was reportedly weaker in male than in female rats [21,26], suggesting a possibility that male gonadal steroids attenuate and/or female gonadal steroids enhance orexin A neuron activation in response to fasting. In the present study, we found that the response of orexin A neurons and neurons in the Arc to reduced glucose availability was stronger in the ORX group than in the Sham group. The data in the present study suggest that the attenuated activation of orexin A neurons during fasting in male rats compared with female rats is possibly due, at least in part, to the attenuated response of orexin A neurons to reduced glucose availability by endogenous androgens.

Feeding behavior was stimulated by 2DG-induced glucoprivation in both the ORX group and the Sham group, while the changes in the plasma glucose concentration and T_abdo_ after 2DG administration were not different between the ORX group and the Sham group. We speculate that the level of glucoprivation, or the level of stimulation, in glucosensitive neurons was similar between the ORX group and the Sham group because the increase in plasma glucose and the decrease in body temperature were not different between these two groups. Therefore, orchiectomy augments, or endogenous androgens, attenuate the response of orexin A neurons to reduced glucose availability.

Animal studies have convincingly demonstrated that androgens increase body weight gain and change body composition. In some studies, orchiectomy reduced food intake, and androgen replacement in orchiectomized animals normalized food intake [1,20,21]. It has also been reported that androgens change the body weight and composition in the absence of changes in food intake [1,9,20]. In the present study, we examined the effect of orchiectomy on spontaneous daily food intake and body weight gain for one week after a one-week recovery period from the surgical stress of orchiectomy/sham operation. We failed to find a difference in daily food intake between the ORX group and the Sham group, although body weight gain after the orchiectomy/sham surgery was greater in the Sham group than in the ORX group. This result is consistent with the results of Chai’s study, in which they reported that orchiectomy reduced daily food intake up to five days after the surgery but did not affect daily food intake thereafter [20]. The data in the present study and earlier studies suggest that androgens affect body weight and body composition by modifying metabolism more prominently than feeding behavior [1,12,13].

Daily spontaneous food intake was not altered by orchiectomy in the present study, while the feeding response and activation of orexin A neurons and neurons in the Arc to reduced glucose availability was augmented by orchiectomy. The orchiectomy-induced augmentation of feeding response to reduced glucose availability did not directly affect daily food intake in the present study. The regulation of feeding behavior is composed of various regulatory systems [1,15], which increase or decrease food intake, and endogenous androgens possibly affect various regulatory systems. Endogenous testosterone reduces satiety, which is induced by elevated plasma glucose concentration after fasting, via the attenuation of MCH neurons’ response to glucose, and increases food intake in response to fasting [21,22]. In contrast, in the present study, we found that feeding induced by reduced glucose availability was attenuated by endogenous androgens. The integrated effect of endogenous androgens on these and other feeding regulatory systems may result in an unchanged daily food intake.

The direct site of action of testosterone, causing diminished food intake and the activation of orexin A neurons and neurons in the Arc, remains unknown in the present study. It is possible that testosterone directly attenuates orexin A neurons in the LH/PFA and neurons in the Arc because androgen receptors are relatively densely distributed in these brain areas [25]. However, androgen receptors were not colocalized with orexin A in neurons of the lateral hypothalamic area, suggesting that androgen’s action in the orexin A neurons is mediated by afferents from neurons containing androgen receptors [31].

Another possible site of action of testosterone is the arcuate nucleus. Kisspeptin neurons in the arcuate nucleus are involved in the negative feedback regulation of the HPG axis and express androgen receptors [32,33]. Kisspeptin has been shown to enhance the expression of orexigenic NPY and attenuate the expression of anorexigenic POMC [32,34,35,36], and the glucosensitive NPY neurons are known to connect reciprocally with orexin A neurons [37,38,39]. These findings suggest that the enhanced responses of feeding and orexin A neuron activity to reduced glucose availability in orchiectomized rats are mediated by increased NPY, induced by a lower level of androgens. The enhanced c-Fos expression in the dorsomedial Arc by orchiectomy in the present study suggests a possibility that reduced androgen enhances the NPY neurons by stimulating the kisspeptin neurons. However, kisspeptin has also been reported to excite POMC neurons and inhibit NPY neurons [32,34,40], and intracerebroventricular kisspeptin reduces food intake [41], suggesting that kisspeptin has anorexigenic properties. In contrast, some studies reported that an intracerebroventricular injection of kisspeptin had no effect on food intake [32,42]. It is expected that additional studies will further elucidate the role of kisspeptin in the regulation of feeding behavior.

The attenuated feeding and activity of orexin A neurons and neurons in the dorsomedial Arc to reduced glucose availability by endogenous androgen might be mediated by estradiol aromatized from testosterone. Relatively high doses of testosterone can be aromatized to estrogens and affect feeding behavior [1], but no exogenous testosterone was administrated in the present study. We speculate that estradiol is not involved in the attenuation of the responses of feeding and orexin A neuron activity to reduced glucose availability by endogenous androgen. However, further research is expected to be carried out to examine the involvement of estradiol.

In the present study, plasma testosterone concentration in the ORX group was relatively high (35% of the sham group). The source of testosterone in the ORX group is unknown, although at least part of it might be secreted from the adrenal glands. One possible reason is that orchiectomy was not completely done in two rats (Figure 2D). However, we consider that the effect of endogenous androgens was able to be clarified because the plasma testosterone level was significantly lower in the ORX, and 2DG-induced food intake was significantly correlated with the plasma testosterone concentration.

## 5. Conclusions

Orchiectomy enhances the feeding response and the activation of orexin A neurons and neurons in the dorsomedial Arc, and food intake induced by reduced glucose availability was inversely correlated with plasma testosterone level. The data indicate that endogenous androgen, possibly testosterone, diminishes the food intake induced by reduced glucose availability, possibly via the attenuated activity of orexin A neurons and neurons in the dorsomedial Arc. However, daily food intake was not altered, and body weight gain was attenuated by orchiectomy. These results suggest that endogenous androgens exert multiple actions on feeding and body weight regulation.

## Figures and Tables

**Figure 1 nutrients-14-01235-f001:**
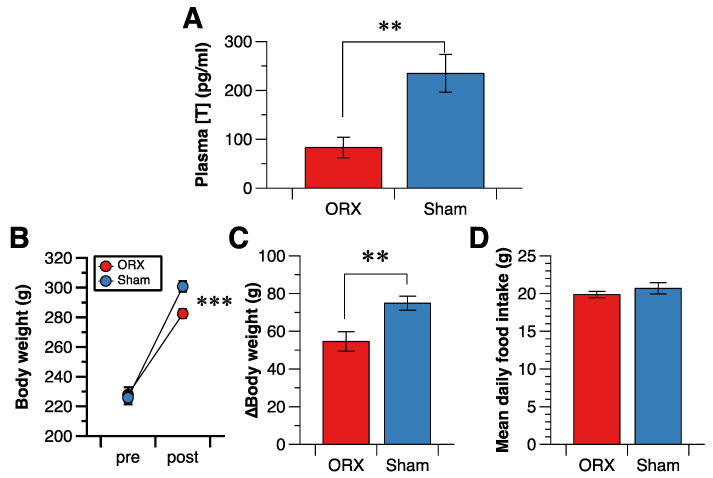
Effect of orchiectomy on plasma testosterone concentration and on body weight and daily food intake in male rats. (**A**): plasma testosterone concentration 21 days after the orchiectomy/sham operation. (**B**): Body weight before (pre) and 14 days after (post) orchiectomy/sham operation. (**C**): Change in body weight from before to 14 days after orchiectomy/sham operation. (**D**): Mean daily food intake for seven days after the 7-day recovery from the orchiectomy/sham operation. Data are shown as the mean and SE (*n* = 11 in each group A, B and C; and *n* = 6 in each group in D). ** and ***: significant difference between the ORX group and the Sham group at *p* < 0.01 and *p* < 0.001, respectively.

**Figure 2 nutrients-14-01235-f002:**
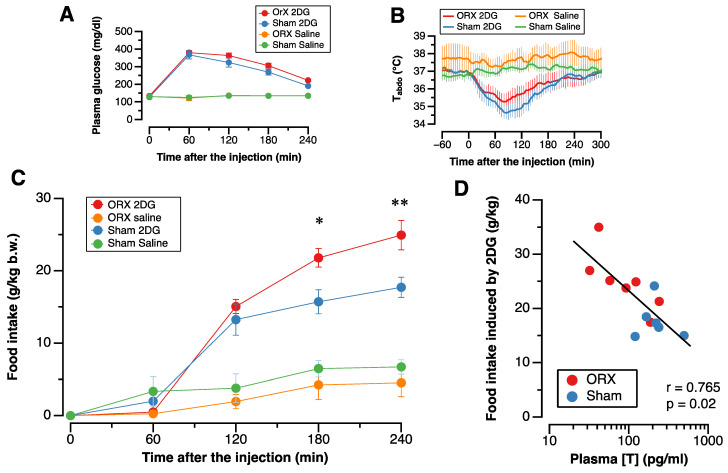
Effect of orchiectomy on responses to inatravenous 2-deoxy-d-glucose administration. (**A**): Plasma glucose concentration, (**B**): Intra-abdominal temperature (T_abdo_). (**C**): Cumulative food intake (**C**) after 2-deoxy-d-glucose (2DG; 400 mg/kg body weight, i.v.) or physiological saline (Sal) administration in the orchiectomized (ORX) and the sham-operated (Sham) groups. Data are shown as mean and SE (*n* = 7 in the ORX_2DG group; *n* = 6 in the Sham_2DG; *n* = 4 in the ORX_Sal; and *n* = 4 in the Sham_Sal group). * and **: significant difference between the ORX_2DG group and the Sham_2DG group at *p* < 0.05 and *p* < 0.01, respectively. (**D**): Relationship between 2-deoxy-d-glucose (2DG)-induced food intake and plasma testosterone concentration ([T]) in the orchiectomized (ORX) group and in the sham-operated (Sham) group. 2DG-induced food intake was significantly inversely correlated with plasma [T] (r = 0.77; *p* = 0.02).

**Figure 3 nutrients-14-01235-f003:**
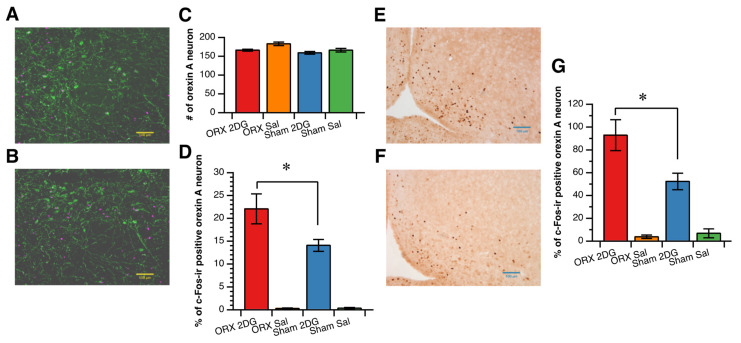
Effect of orchiectomy on the activation of orexin A neurons (**A**–**D**) and the activation of neurons in the arcuate nucleus (**E**–**G**) induced by 2-deoxy-d-glucose (2DG) administration. (**A**,**B**): Representative microscope images of c-Fos-immunoreactive (ir) nuclei (magenta) and orexin A immunoreactivity (green) in the lateral hypothalamic/perifornical area in the orchiectomized (ORX) group (**A**) and in the sham-operated (Sham) group (**B**). Scale bar = 100 µm. (**C**): Number of orexin A neurons in the lateral hypothalamic/perifornical area (917 × 662 µm). (**D**): Percentage of c-Fos-ir-positive orexin A neurons. (**D**,**E**): Representative microscope images of c-Fos-immunoreactive (ir) nuclei (dark brown) in the arcuate nucleus in the orchiectomized (ORX) group (**E**) and in the sham-operated (Sham) group (**F**). Scale bar = 100 µm. (**G**): Number of c-Fos-ir neurons in the arcuate nucleus. Data are shown as the mean and SE in C and D (*n* = 7 in the ORX_2DG group; *n* = 6 in the Sham_2DG; *n* = 4 in the ORX_Sal; and *n* = 4 in the Sham_Sal group). *: significant difference between the ORX_2DG group and the Sham_2DG group at *p* < 0.05.

## Data Availability

The data will be available on request.

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
