# Peer review of "Endogenous Androgens Diminish Food Intake and Activation of Orexin A Neurons in Response to Reduced Glucose Availability in Male Rats"

_nutrients, 2022, doi:10.3390/nu14061235_

Round 1
Reviewer 1 Report
I thank the journal for the opportunity to review this interesting study. The topic is of great interest, although quite controversial. The paper is well structured and interesting; however, I have some comments that, in my opinion, should be clarified in a revised version.
ABSTRACT:
It is well structured and informative. I think clearly describe the study
INTRODUCTION:
I apologize for my ignorance, buy andropause and hypogonadism are synonyms? (1st paragraph p2).
In line 67, again, I am not sure if authors may refer to satiety or hunger? That is, testosterone reduces satiety, namely increase hunger? or is it just the opposite?
I miss some studies conducted on humans to better interpret the relevance of the present paper.
Also, I expect more controversy regarding the role of orchiectomy on body fat and, in special, body composition. What is the experience on pets?
MATERIALS AND METHODS
Is it the paper adhered to any international standard guidelines like ARRIVE for animal handling?
Line 124. How were the blood samples obtained? What volume?
I think section 2.4 is unnecessary and only describe standard methods.
In the statistics section, I miss the rationale for analyzing such a small sample. Have you calculated the observed power? In my experience, I have never obtained a size of n=4 in two-way ANOVA. What previous studies were you based on to calculate d or sd?
RESULTS
In the ORX group, testosterone levels were of (approx.) 100 pg/ml. What was the origin of that testosterone? Again, I regret my ignorance in this regard. I understand that the adrenal glands can synthesize a little testosterone (about 10% in man), but in rats, do they produce about 50%? Looking at figure 1, that seems to be the difference between ORX and sham groups.
The statistical analysis in figure 4 is, at least, questionable. The correlation in ORX group seems evident, but in the sham group, there is an evident lack of correlation. I am not sure about the idoneity of combining animals of both groups for this analysis. In addition, I am not sure about correlation analysis with n=6+7.
In addition, the data of figure 4 contrast with the data of figure 2C and 2D. That is, attending to figure 4, the higher testosterone the lower food intake induced by DG (at short-term). However, at long term, the lower testosterone the lower body weight (as seen in figure 2B). Moreover, in figure 2C ORX animals had a lower intake than sham animals. This contradiction between 2DG induced and long-term appetite ratings have not been clarified throughout the study (in my opinion).
DISCUSSION
Lines 284-286. Again, you commented that "endogenous androgens diminish the short-term feeding behaviour", however, ORX rats have lower body weight. This contradiction should be clarified at the beggining of the discussion.
In this regard, discussion is evidently focused on food intake and body weight, but this brings a lot of controversy between short and long-term data. I believe that having analyzed body composition, especially regarding the amount of water and especially fat, could greatly explain part of the results obtained.
The section about kisspeptin neurons is quite speculative and out of the results obtained. I will consider to remove the entire paragraph.
What is the reason for not share the data?
Reviewer 2 Report
The manuscript titled “Endogenous Androgens Diminish Food Intake and Activation of Orexin A neurons in the Lateral Hypothalmic/Perifonical Area in response to Glucoprivation in Male Rats” determine the effect of orchiectomy on 2DG induced glucoprivation stimulated feeding behavior and orexin A neurons activity. The manuscript has many short comings, in-depth analysis of the study needs to be done. The manuscript itself is poorly written and organized, many grammatical mistakes need to be corrected. Following are some of my point-by-point comments but the manuscript needs further experimentation, reorganization of both the data and text.
- The abstract and introduction should be written in simplified language, the authors should explain the terms like orchiectomy, glucoprivation, and outcome of the neuronal function in a specific region of the brain instead of just the names such as the function of orexin, arcuate nucleus, LH/PFA and so on, to make it easier to understand for the reader.
- Figures 1 and 2 should be merged as it is part of the same experiment.
- Figure 2 showed a reduced change in body weight in ORX mice without change in daily food intake. To explain this observation, authors should further determine the cause of differential weight gain; is it the change in energy expenditure, change in lean/fat mass etc. Further anlysis is needed to explore the mechanism of orchiectomy on the metabolic cycle.
- Similarly, Figures 3 and 4 will combine to become figure 2.
- In section 3.3, the authors measured c-Fos expression in orexin A neurons and in the Arc. The authors should explain the background for measuring c-Fos expression, it's downstream signaling, and hence its effect on food intake. Similarly, Figures 5 and 6 will combine to become figure 3.
- In addition to measuring c-Fos, authors should do more experiments to validate the activity of orexin A and how androgens play a role in decreasing its activity. They should show the in vitro or in vivo mechanism of the effect of androgens on orexin A, as this was the main aim of the manuscript. Current data only supports correlation, without mechanistic insight. Thus, further experimentation is strongly recommended.
- Please check the grammar thoroughly throughout the manuscript.
The need of the study explained by the authors is “to elucidate the role of endogenous androgen in the short-term regulation of feeding behavior”. However, the experimental design is very superficial, the in-depth mechanistic study on the effect of androgen on suppression of orexin A neuron is required to publish this manuscript.
Round 2
Reviewer 2 Report
Dear Authors,
I appreciate your response to the reviewer's comment. In the light of your phenomenological finding, I would suggest publishing it as a letter instead of a full research article.